# A developmental gradient of COUP-TFI expression regulates the relative size of hippocampus dorsal and ventral subregions

Ching-San Tseng[1,2], Zi-hui Zhuang[1], Hsiang-Wei Hsing[1], Chia-Ming Lee[3], Tsan-Ting Hsu[4], Yu-Kuan Pan[1], Yi-Ping Hsueh[4], Bi-Chang Chen[3], Shen-Ju Chou[1]*

1 Institute of Cellular and Organismic Biology, Academia Sinica, Taipei, Taiwan, 2 Department of Anatomy, School of Medicine, China Medical University, Taichung, Taiwan, 3 Research Center for Applied Sciences, Academia Sinica, Taipei, Taiwan, 4 Institute of Molecular Biology, Academia Sinica, Taipei, Taiwan

* schou@gate.sinica.edu.tw

## Abstract

The hippocampus is a key brain region for learning and memory that can be divided into multiple compartments, including the dorsal and ventral hippocampi. Each compartment has unique functions, connectivity, and molecular properties. However, the molecular mechanisms underlying the specification of these subregions remain unclear. In this study, we investigated the role of COUP-TFI, a patterning transcription factor expressed in a high-caudal-to-low-rostral gradient in hippocampal progenitors, in establishing the dorsal-ventral axis of the hippocampus in mice. The highest COUP-TFI expression in the hippocampal progenitors was observed in the caudal medial pallium that gives rise to the ventral hippocampus, and the lowest expression was found in rostral medial pallium that develops into the dorsal hippocampus. We generated mice with different levels of COUP-TFI expression in the cortex and performed a comprehensive analysis of subregion-specific gene expression across the entire hippocampus in these mice. The deletion of COUP-TFI resulted in an expansion of the dorsal hippocampus at the expense of the ventral hippocampus. Conversely, increased COUP-TFI expression led to the generation of ectopic ventral hippocampus in the dorsal compartments. We further demonstrated that COUP-TFI dose-dependently regulates Wnt signaling pathway, a crucial player in regulating hippocampus development. Together, our findings reveal a novel mechanism governing hippocampal development, in which COUP-TFI determines the fate of hippocampal subregions and balances the hippocampus into dorsal and ventral compartments.

## Introduction

The hippocampus (Hp) is a major learning and memory center in the limbic system that can be divided into multiple domains. Along the rostrocaudal axis, at least two distinct functional domains, the dorsal and ventral Hp, have been identified. The

**Data availability statement:** All relevant data are within the paper and its Supporting Information files.

**Funding:** This work was supported by the National Science and Technology Council, Taiwan (https://www.nstc.gov.tw; NSTC 113-2811-B-001-066 and NSTC 113-2311-B-001-004, S.-J.C.) and Academia Sinica (https://www.sinica.edu.tw; AS-TP-110-L10, and the Institute of Cellular and Organismic Biology, S.-J.C.). The funders had no role in study design, data collection and analysis, decision to publish, or preparation of the manuscript.

**Competing interests:** I have read the journal's policy and the authors of this manuscript have the following competing interests: Y.-P.H. is a member of PLOS Biology's Editorial Board. The other authors declare that no competing interests exist.

dorsal Hp (dHp), which corresponds to the posterior Hp in primates, is involved in spatial learning. Place cells (neurons that fire in response to specific spatial information) are enriched in the dHp [1,2]. Ablation or defective lamination of the dHp disrupts spatial memory [3–5]. However, ablation of the ventral Hp (vHp), corresponding to the anterior Hp in primates, does not affect spatial learning. Instead, vHp ablation or dysfunctions significantly reduced anxiety-like behaviors [6–8]. Anxiety cells (neurons that fire during anxiogenic situations) have been identified in vHp [9]. Supporting these functional differences, distinct Hp domains, including the dHp, vHp, and the intermediate Hp (positioned between the dorsal and ventral Hp), have unique gene expression profiles, cytoarchitectures, neuronal excitability, and connectivity patterns [10–17]. In this study, we investigated how these subregions are specified during development.

During corticogenesis, the Hp originates from the hippocampal neuroepithelium (HNE) in the medial pallium [18,19]. Signaling molecules, such as Wnts expressed in the cortical hem (CH), and transcription factors (TFs), like Lhx2 and Emx2 expressed in the HNE, are known to play critical roles in the specification and growth of the Hp [20,21]. Within HNE, COUP-TFI (also known as NR2F1) is expressed in a high-caudal-to-low-rostral gradient. Recent studies showing that the deletion of COUP-TFI resulted in a dysmorphic Hp with a specific loss of the dHp [22,23] prompted us to study whether COUP-TFI expression gradient is involved in patterning Hp subregions. However, to our surprise, different from what was reported based on marker gene analyses done on limited numbers of sections, our systematic examination of multiple subregion-specific markers across the entire hippocampus at postnatal day 7 (P7) revealed that in the COUP-TFI conditional knockout (cKO) mice, where COUP-TFI is deleted in the cortex by the Emx1-Cre, the relative size of dHp was actually expanded at the expense of the vHp. To further investigate this issue, we analyzed Hp patterning in the COUP-TFI conditional transgenic (cTG) mice, where COUP-TFI expression is elevated in the cortical progenitors, including the HNE [24]. We found that increased COUP-TFI expression led to a ventralized Hp in which vHp invaded the dorsal regions forming ectopic ventral structures. These patterning changes in the cKO and cTG were consistently observed in the Cornu Ammonis (CA)1 and CA3 regions. Our findings suggested that the high expression level of COUP-TFI promotes the vHp identity, while loss of COUP-TFI shifts the balance toward dHp fate. Additionally, the formation of ectopic vHp structures in the dorsal territories in the cTG suggests that the Hp subregions have differential cell affinities, which are likely involved in compartmental segregation.

Mechanistically, our results showed that the low-rostral-to-high-caudal gradient of COUP-TFI in the HNE is involved in generating the expression gradient of several downstream genes in the Wnt signaling pathway. Thus, we conclude that COUP-TFI, together with Wnt signaling, is crucial for patterning Hp into distinct functional domains along the dorsoventral axis to ensure proper hippocampal functions.

## Results

### COUP-TFI expression level regulates hippocampal size

To determine the overall structure and size of the Hp, which is embedded in the cortex, we employed light sheet microscopy on cleared brains of Thy1-YFP transgenic mice (Thy1-YFP-H line) [25,26]. Consistent with previous reports, we observed an uneven distribution of YFP-expressing neurons across various brain regions, with higher densities in the deep layer of the motor area of the frontal cortex (F/M), auditory cortex (AuD), nucleus accumbens (Acb), amygdala (Amg), and Hp [27] (Fig 1A and S1 Movie). We further noted that the density of YFP-expressing neurons changed along

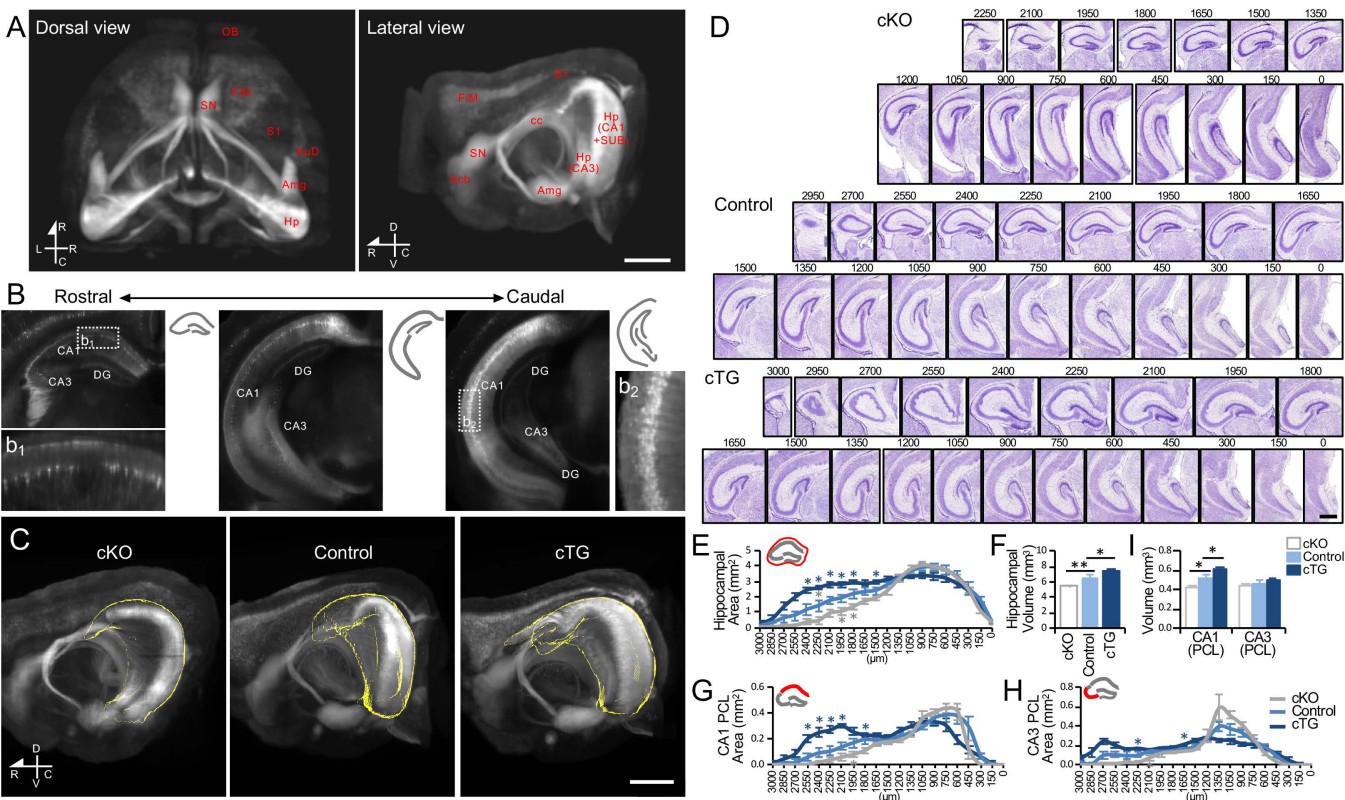

**Fig 1. Hippocampal volume changes along the rostrocaudal axis in the COUP-TFI mutant mice. (A)** 3D whole brain imaging of Thy1-YFP expression in the adult control cortex. Left, dorsal view; right, lateral view. **(B)** Distribution of YFP+ cells in the hippocampus along the rostrocaudal axis. Sparse YFP+ cells were found in the rostral CA1 (b₁). The density of YFP+ cells increased along the rostrocaudal axis. A dense distribution of YFP+ cells was found in the caudal CA1 (b₂). **(C)** Changes in YFP+ cell distribution in the COUP-TFI mutant hippocampus (marked by yellow outline). The size of the domain with dense YFP+ cells in the caudal hippocampus was decreased in the COUP-TFI cKO (*COUP-TFI*f/f; *Emx1*-Cre) and increased in the COUP-TFI cTG (*COUP-TFI*TG/O; *Emx1*-Cre). **(D)** Serial Nissl-stained coronal sections containing hippocampus (Hp) from P7 COUP-TFI cKO, control, and COUP-TFI cTG cortices. The numbers at the top indicate the distance (μm) of each displayed section from the caudal end of the hippocampus (position 0). **(E)** The overall hippocampal area along the rostrocaudal axis in control, COUP-TFI cKO, and cTG mice. The distance of the displayed section from the caudal end of the Hp is indicated. In the rostral sections (1,800–3,000 μm), the Hp area size is significantly smaller in the cKO and larger in the cTG. **(F)** Estimated Hp volumes in control, COUP-TFI cTG, and cKO mice. Significantly smaller and larger Hp were observed in COUP-TFI cKO and cTG mice, respectively. **(G, H)** The area of the pyramidal cell layer (PCL) area in the CA1 (d) and CA3 (e) regions along the rostrocaudal axis in control, COUP-TFI cKO, and cTG mice. **(I)** Estimated volumes of CA1 and CA3 PCL in control, COUP-TFI cKO, and cTG mice. Significant changes were found in COUP-TFI mutants for CA1 but not CA3. Scale bars, 1 mm (A, C), 500 μm (B), 200 μm (D). Acb, accumbens nucleus; AuD, auditory cortex; Amg, amygdala; F/M, motor area in frontal cortex; Hp, hippocampus; OB, olfactory bulb; S1, primary somatosensory area; SN, septal nuclei. Statistical analyses were performed using Mann–Whitney *U* test (E, G, H) and Student *t* test (F, I). The data underlying this figure can be found in S1 Data.

the rostrocaudal axis in Hp. Along with the higher YFP-expressing cell densities in the caudal part of CA1 and CA3, YFP signals were noticeably stronger in the caudal Hp (Fig 1B).

To determine whether COUP-TFI expression level affects the development and patterning of the Hp, we crossed Thy1-YFP transgenic mice with COUP-TFI cKO (COUP-TFI$^{f/f}$; Emx1$^{Cre/+}$) and cTG (COUP-TFI$^{TG/O}$; Emx1$^{Cre/+}$) mouse lines, which respectively eliminated and elevated COUP-TFI expression in the cortex during cortical development. The overall cortical sizes were similar between control mice (COUP-TFI$^{f/f}$, or COUP-TFI$^{TG/O}$) and COUP-TFI mutants, including cKO and cTG at 2 months of age and at P7 (Figs 1C and S1). Notably, COUP-TFI cKO mice exhibited enlarged ventricles with a smaller Hp than the control brains, agreeing with previous findings [22,23] (Fig 1C). Correspondingly, the Hp was larger in the cTG. We found the YFP expression level in the Hp was decreased in the cKO, while the domain with strong YFP signals expanded in the cTG (Fig 1C and S1 Movie). These findings suggest that COUP-TFI plays a crucial role in regulating Hp size and patterning.

To confirm the size changes in the Hp, we estimated the overall size of the Hp and the individual sizes of Hp subregions on Nissl-stained serial coronal sections from P7 control and COUP-TFI mutant cortices. Compared with the control cortex, the rostral end of the Hp appeared in a more caudal section of the cKO and in a more rostral section of the cTG (Figs 1D and S2A), meaning that the overall Hp size was significantly reduced in the cKO (n = 8; $p = 0.003$) and increased in the cTG (n = 8; $p = 0.027$) (Fig 1E and 1F). We further compared the areas of pyramidal cell layers (PCLs) of CA1 and CA3 regions between control and COUP-TFI mutants. In rostral sections, we found that the PCL areas in both CA1 and CA3 were smaller in the cKO, and larger in the cTG (Fig 1G and 1H). Although the overall size of CA3 remained similar between control and COUP-TFI mutants, CA1 size was significantly decreased in cKO mutants and increased in cTG mutants (control versus cKO, $p = 0.013$; control versus cTG, $p = 0.034$) (Fig 1I). Thus, we showed that Hp volume correlates with COUP-TFI expression level; a smaller Hp was observed in cKO mice with lower COUP-TFI expression, while a bigger Hp was found in cTG with higher COUP-TFI expression.

Importantly, the changes in Hp size in the COUP-TFI mutants were not due to differences in embryonic neurogenesis. The distributions and densities of neurons labeled with EdU pulses at E13.5 and E15.5 were comparable between control and COUP-TFI mutants in both CA1 and CA3 PCLs at P0 (S2B and S2C Fig). Furthermore, the inside-out neurogenetic gradient in the CA1 region was maintained in the COUP-TFI mutants in both rostral and caudal levels, as evidenced by a relatively even distribution of E13.5-born neurons and an enrichment of E15.5-born neurons in superficial sublayers in both rostral and caudal sections [28] (S2D Fig). In contrast, EdU labeled neurons in CA3 region did not exhibit a clear laminar distribution at P0 in either control or mutant animals (S2E Fig).

## COUP-TFI expression level regulates dorsal-ventral balance of CA1

In addition to the changes in size, we observed structural changes in the COUP-TFI mutant Hp at P7. As CA1 PCL thickness and cell density changed along the rostrocaudal axis in the control Hp (Figs 2A and S3A): the PCL in dorsal CA1 (dCA1) was significantly thinner and had a higher cell density compared to that in ventral CA1 (vCA1) (n = 8; thickness, $p < 0.001$; cell density, $p < 0.001$) (Fig 2A–C). We selected a rostral and a caudal section to compare among different genotypes based on their similarity in Hp morphology and relative locations (as shown in Fig 2A). We found the CA1 PCL thickness was significantly decreased in cKO (n = 8; rostral, $p = 0.010$; caudal, $p = 0.008$) and significantly increased in the cTG (n = 8; rostral, $p = 0.009$; caudal, $p = 0.024$) at both levels (Fig 2A and 2B). Furthermore, the CA1 PCL cell density was significantly increased in the cKO (n = 7–8; rostral, $p = 0.045$; caudal, $p = 0.005$) and decreased in the cTG (n = 7–8; rostral, $p = 0.041$; caudal, $p = 0.027$) (Fig 2C). Additionally, except that the stratum radiatum (SR) was thinner in the cKO CA1, we did not observe significant differences in thickness and cell density of the SR and the stratum oriens (SO) in control and COUP-TFI mutants (S3A and S3B Fig).

We then measured the thickness of PCL in the middle of CA1 over the entire rostrocaudal axis and showed the increase of the CA1 thickness from rostral to caudal in both control and COUP-TF mutant mice (S3C Fig). Furthermore,

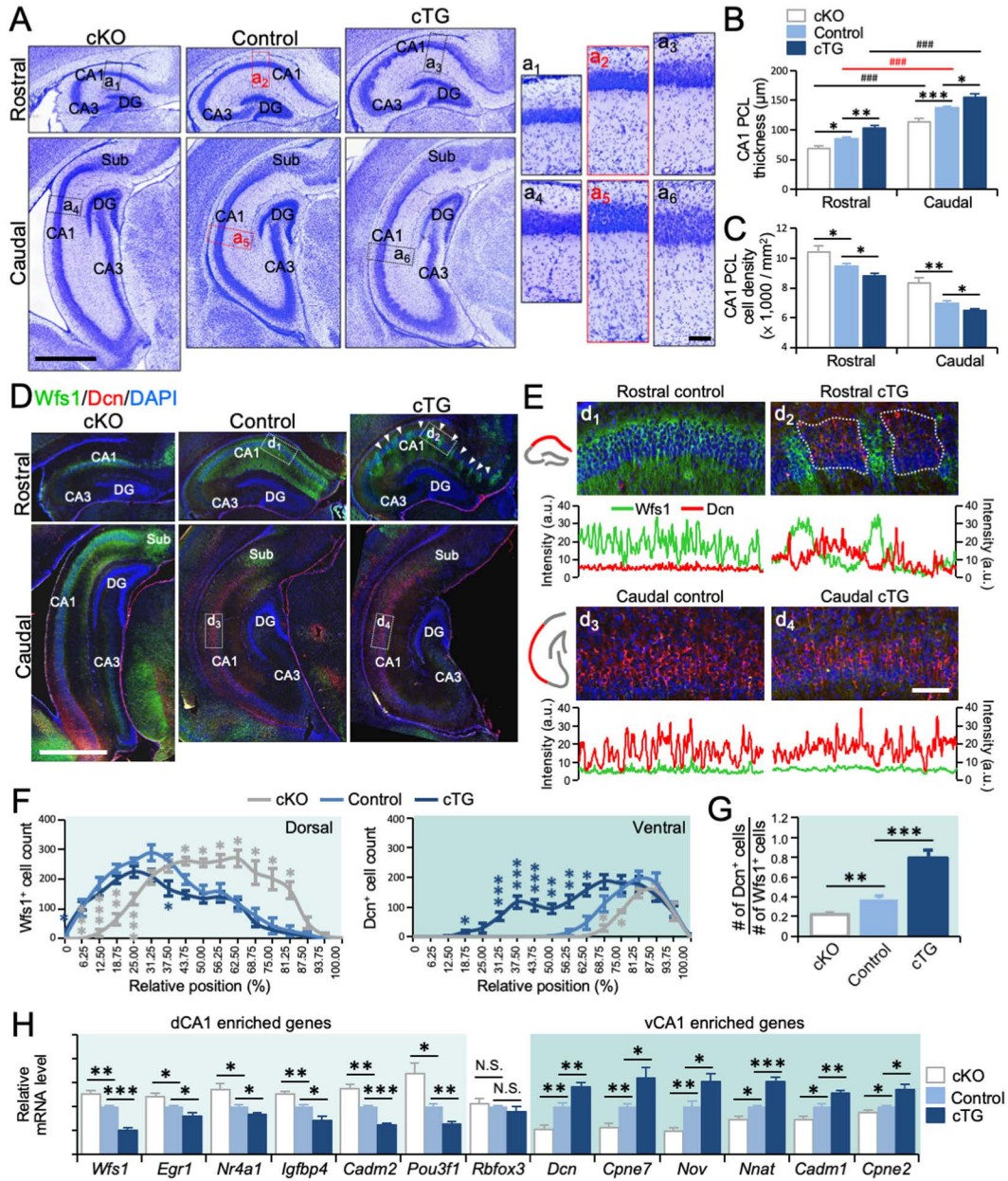

**Fig 2. Changes in dorsal-ventral patterning of CA1 in COUP-TFI mutant hippocampus. (A)** Nissl-stained coronal sections of P7 COUP-TFI cKO, control, and COUP-TFI cTG at rostral and caudal levels. The right panels show higher magnifications of insets ($a_1$-$a_6$). **(B, C)** Quantification of layer thickness (B) and cell densities (C) in CA1 PCL in COUP-TFI cKO, control, and COUP-TFI cTG hippocampus at rostral and caudal levels. **(D)** Immunostaining for Wfs1 (green) and Dcn (red) on P7 coronal sections of COUP-TFI cKO, control, and COUP-TFI cTG Hp at rostral and caudal levels. The Wfs1 expression became fragmented in the rostral CA1 of cTG, with Wfs1- domains (arrowheads). **(E)** The expression domains of Wfs1 and Dcn were restricted to the rostral and caudal CA1, respectively, in the control. Ectopic Dcn-expressing cell clusters were found next to Wfs1-expressing cells in the rostral CA1 in COUP-TFI cTG. **(F)** The number of Wfs1+ and Dcn+ cells along rostrocaudal axis in COUP-TFI cKO, control, and COUP-TFI cTG Hp. **(G)** Ratio of total number of Dcn+ and Wfs1+ cells in CA1 regions of control, COUP-TFI cKO, and cTG. The Dcn/Wfs1 ratio was significantly decreased in the cKO and increased in the cTG. **(H)** Relative expression levels of genes enriched in dorsal and ventral CA1 in control, COUP-TFI cKO, and cTG Hp. Scale bars, 1 mm (A, D); 100 μm ($a_1$-$a_6$, $e_1$-$e_4$). DG, dentate gyrus; Sub, subiculum. Statistical analyses were performed using Mann–Whitney $U$ test (F) and Student $t$ test (B, C, G, H). The data underlying this figure can be found in S1 Data.

the PCL was thinner in cKO CA1, especially in the rostral sections, and thicker over the entire CA1 region in cTG (S3C Fig). To ensure that the Hp cytoarchitectural and molecular properties were compared at the same rostrocaudal levels across genotypes, we scaled up the smaller Hp in cKO and scaled down the larger Hp in cTG to align the most rostral ends and most caudal ends of Hp among all samples. Using this analysis, we still found that CA1 PCL was significantly thinner in the cKO and thicker in the cTG at correlated positions along the rostrocaudal axis (S3C' Fig).

Using the CA1 pyramidal cell marker Satb2 and Ctip2, we found that in control Hp, most of the CA1 pyramidal neurons co-expressed Satb2 and Ctip2 (Satb2$^+$ Ctip2$^+$), whereas a population expressing only Ctip2 (Satb2$^-$ Ctip2$^+$) was enriched in the deep sublayer in caudal CA1 PCL (S3D and S3E Fig). The change in neuronal composition correlated with the changes in CA1 PCL laminar thickness. We found the caudal-enriched Satb2$^-$ Ctip2$^+$ population was reduced in the caudal CA1 in cKO, but increased in both rostral and caudal CA1 in cTG (S3D and S3E Fig). Together with the reduced thickness and increased cell density in the cKO CA1 PCL and the increased thickness and decreased cell density in the cTG CA1 PLC, these findings suggested that the Hp was dorsalized in cKO and ventralized in cTG. This finding agreed with the Thy1-YFP expression patterns observed in the COUP-TFI mutant Hp, but it challenged previous reports suggesting that the dorsal Hp failed to form in cKO mutants [22,23].

To further determine the Hp patterning changes induced by altering COUP-TFI expression levels, we stained the Hp tissues with dHp- and vHp- specific markers in COUP-TFI mutants at P7. We used two mutually exclusive markers, Wolframin (Wfs1) to label the rostrally located dCA1 and Decorin (Dcn) to label the caudally located vCA1 [12–15,29,30] (Figs 2D, S3C, and S9A). In the cKO, Wfs1 was expressed in rostral CA1, similar to the control, but its expression domain was expanded to the caudal CA1. Moreover, the Dcn$^+$ domain in the cKO Hp was shifted ventrally in the caudal CA1 (Fig 2D). In the cTG, while Dcn expression patterns in the vCA1 remained similar to that in the control, we found Wfs1 expression domain in the dCA1 became fragmented (Fig 2D). In some of the Wfs1$^-$ domains, we detected ectopic expression of Dcn (Figs 2E and S4), suggesting the formation of ectopic vCA1 within the dCA1 in cTG. However, some Wfs1$^-$ domains, especially in more rostral sections, did not express Dcn (S4 Fig). To quantify the changes in the balance between dorsal and ventral Hp in COUP-TFI mutants, we aligned the rostral and caudal ends of the Hp among all genotypes as described above and compared Wfs1$^+$ and Dcn$^+$ cell numbers at comparable levels along the rostrocaudal axis between control and COUP-TFI-mutant Hp. This analysis confirmed that the Wfs1$^+$ dCA1 expanded and the Dcn$^+$ vCA1 shifted caudally in the cKO (Fig 2F). As such, the ratio of total Dcn$^+$ vCA1 cells to Wfs1$^+$ dCA1 cells in the entire Hp was significantly decreased in the cKO compared with control (n = 7–8; $p = 0.043$) (Fig 2G), suggesting the cKO CA1 was dorsalized. Meanwhile, the fragmented Wfs1$^+$ domain and ectopic Dcn expression in the cTG dorsal CA1 caused a significant increase in the ratio of Dcn$^+$ to Wfs$^+$ cells in the cTG (n = 8; $p = 0.004$) (Fig 2F, 2G), suggesting ventralization of the CA1.

We further considered the possibility that hippocampal curvature might vary between genotypes. To address this, we performed immunostaining for Wfs1 and Dcn on transverse sections spanning from the dorsal to the ventral pole of flattened hippocampi from different genotypes (S5A Fig). With this, we confirmed that Wfs1 expression domain was expanded while Dcn expression domain was reduced in cKO Hp. Furthermore, in cTG Hp, Wfs1 expression in the dorsal CA1 also appeared fragmented, accompanied by ectopic domains of Dcn expression (S5B Fig).

We further analyzed a scRNA-seq database [30] to identify additional specific molecular markers for dCA1 and vCA1 (S6 Fig). Measuring total RNA from the entire Hp of control and cKO, we found, while the expression level of *Rbfox3* (the gene encodes NeuN) remained unchanged between control and COUP-TFI mutants, significant increases in the expression of dCA1-enriched genes (n = 6; *Wfs1*, $p = 0.006$; *Egr1*, $p = 0.033$; *Nr4a1*, $p = 0.038$; *Igfbp4*, $p = 0.003$; *Cadm2*, $p = 0.003$; *Pou3f1*, $p = 0.018$) and alongside significant decreases in the expression of vCA1-enriched genes in the cKO (*Dcn*, $p = 0.002$; *Cpne7*, $p = 0.002$; *Nov*, $p = 0.002$; *Nnat*, $p = 0.006$; *Cadm1*, $p = 0.011$; *Cpne2*, $p = 0.044$) (Fig 2H). Correspondingly, significant decreases in the expression of dCA1-enriched genes (n = 6; *Wfs1*, $p < 0.001$; *Egr1*, $p = 0.016$; *Nr4a1*, $p = 0.013$; *Igfbp4*, $p = 0.015$; *Cadm2*, $p < 0.001$; *Pou3f1*, $p = 0.002$) and significant increases in the expression of vCA1-enriched genes were found in the cTG (*Dcn*, $p = 0.008$ *Nov*, $p = 0.025$; *Cpne7*, $p = 0.024$; *Nnat*, $p < 0.001$; *Cadm1*, $p = 0.004$;

*Cpne2*, *p* = 0.014) (Fig 2H). We further validated these findings using in situ hybridization for *Dcn, Cpen7,* and *Nnat*, three genes enriched in the vCA1, to confirm the caudal shift of vCA1 in the cKO and the formation of ectopic vCA1 clusters in the cTG dorsal CA1 (S7 Fig). Thus, we concluded that COUP-TFI regulates the dorsal-ventral patterning of the CA1, with its deletion leading to an expansion of dCA1 and its overexpression leading to the formation of ectopic vCA1 domains.

## COUP-TFI expression levels regulate the position of CA2 and the dorsal-ventral balance of CA3

Given the high interconnectivity among CA1, CA2, and CA3, we next examined whether the patterning changes could be detected in CA2 and CA3 in COUP-TFI mutants at P7. We used Pcp4 (Purkinje cell protein 4) to label CA2 neurons that are located between CA1 and CA3 in the rostral Hp [31] (Fig 3A). In the cKO, a Pcp4$^+$ domain was detected between CA1 and CA3, similar to the control (Fig 3A and S9C). However, we found ectopic Pcp4$^+$ domains in the rostral CA1 of cTG (Fig 3A and 3a2). These ectopic Pcp4$^+$ cells typically clustered in the rostral CA1 of cTG and they did not express Satb2, a general CA1 marker (Figs 3A, S8A, and S9C). The ectopic Pcp4 expression in the cTG dCA1 was accompanied by another CA2 marker, Rgs14 (Regulator of G-protein signaling 14) (S8B Fig). These ectopic CA2 and vCA1 domains contributed to the fragmented dCA1 in cTG Hp and they were typically located more rostrally than the ectopic vCA1 domains (S9C Fig). Furthermore, additional Pcp4$^+$ cells could be detected in the deep layer of CA1 in the caudal Hp (Fig 3a3). In the caudal CA1 of cTG, we also observed an increased number of Pcp4$^+$ neurons compared with the control (Fig 3a4).

Based on the expression of genes specific to dCA3 and vCA3 (S6C Fig), we found CA3 was also dorsalized in the cKO and ventralized in the cTG, similar to our observations in CA1. The dCA3-enriched genes *Prkcd, Cpne9,* and *Trps1* were significantly up-regulated in the cKO (n = 6; *Prkcd*, *p* = 0.048; *Cpne9*, *p* = 0.049; *Trps1*, *p* = 0.045; *Neurod6*, *p* = 0.036; *Ephb1*, *p* = 0.009, *Cyp26b1*, *p* = 0.010) and down-regulated in the cTG (n = 6; *Prkcd*, *p* < 0.001; *Cpne9*, *p* = 0.009; *Trps1*, *p* = 0.015; *Neurod6*, *p* = 0.015; *Ephb1*, *p* = 0.013, *Cyp26b1*, *p* = 0.002) Hp. Conversely, vCA3-enriched genes *Calb2, Plcxd3,* and *Prss23* were significantly down-regulated in the cKO (n = 6; *Calb2*, *p* = 0.025; *Prss23*, *p* = 0.005; *Plcxd3*, *p* = 0.036; *Igfbp3*, *p* = 0.038; *Htr2c*, *p* = 0.036; *Plagl1*, *p* = 0.042) and up-regulated in the cTG (n = 6; *Calb2*, *p* = 0.011; *Plcxd3*, *p* = 0.019; *Prss23*, *p* = 0.012; *Igfbp3*, *p* = 0.002; *Htr2c*, *p* = 0.002; *Plagl1*, *p* = 0.049) Hp (Fig 3B). We confirmed these findings by demonstrating the expression domain of Prkcd (Protein kinase c delta, enriched in dCA3) extended to more caudal sections, while the Calb2 expression (Calretinin, enriched in vCA3) [14] domain was decreased in size in the cKO (Figs 3C and S9B). In cTG, Prkcd expression in dCA3 became fragmented in the rostral sections with ectopic Calb2 expression domains (Figs 3C and S9B). These results suggested that the expression level of COUP-TFI coordinates the dorsal-ventral patterning of multiple CA subregions.

## COUP-TFI expression levels regulate Hp patterning

Using the markers for specific Hp subregions across the entire rostrocaudal axis: Wfs1 for dCA1, Dcn for vCA1, Prkcd for dCA3, Calb2 for vCA3, and Pcp4 for CA2 (S9A–C Fig), we summarized the changes in Hp patterning in COUP-TFI mutant. We rescaled the Hp size in different genotypes to directly compare the relative positions and size of Hp subregions within the Hp. In the cKO, the dCA1 and dCA3 were expanded and caudally shifted, which was associated with relatively smaller and caudally shifted vCA1 and vCA3, as compared to the control (Fig 3D). Conversely, in the cTG, the vCA1 and vCA3 expanded to form ectopic vCA1 and vCA3 domains within the dCA1 and dCA3, respectively (Fig 3D). Additionally, ectopic CA2 domains emerged within the cTG rostral dCA1 (Fig 3D). Thus, we concluded that COUP-TFI expression levels determine the location and size of Hp subregions.

## COUP-TFI regulates Wnt downstream genes to pattern hippocampal CA subfields

To address whether COUP-TFI regulates Hp patterning in hippocampal progenitors or neurons, we crossed COUP-TFI transgenic mice with Nex-Cre mice to overexpress COUP-TFI in postmitotic neurons [32]. In the P7 cTG-Nex

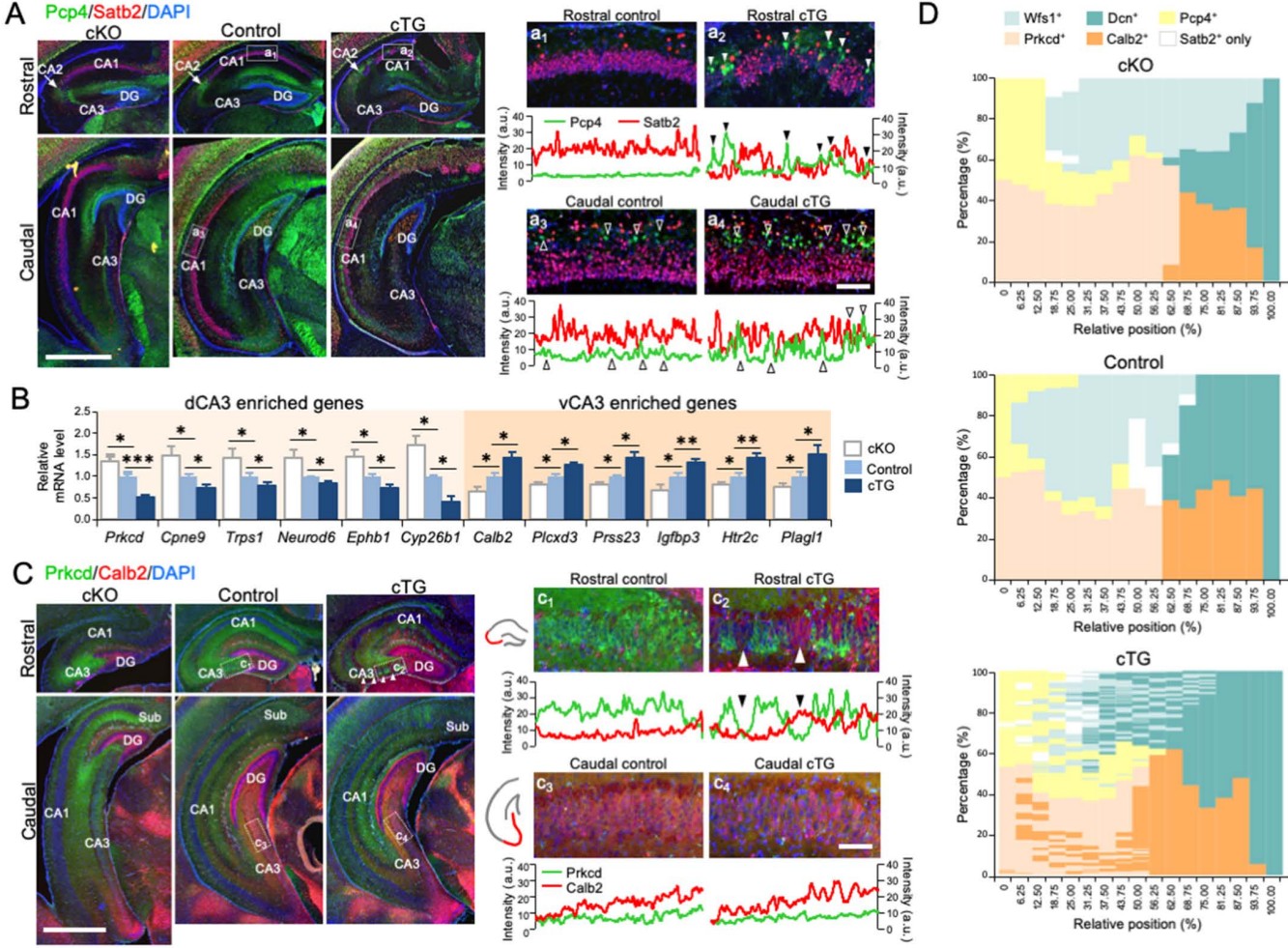

**Fig 3. Hippocampal patterning changes in COUP-TFI mutants. (A)** Ectopic distribution of Pcp4+ neurons in COUP-TFI cTG. Immunostaining of Pcp4 (green) for CA2 (solid arrowheads) and Satb2 (red) for CA1 on P7 coronal sections of COUP-TFI cKO, control, and COUP-TFI cTG Hp at rostral and caudal levels. (a₁-a₄) Higher magnifications of insets in (a). In control, Pcp4 expression was not detected in the rostral CA1, but a few Pcp4+ neurons (open arrowheads) could be detected in the deep layer in caudal CA1. In COUP-TFI cTG, ectopic Pcp4+ neuronal clusters (solid arrowheads) could be found in the Satb2- domains in rostral CA1. Increased numbers of Pcp4+ neurons (open arrowheads) were found in the deep layers of caudal CA1 PCL in COUP-TFI cTG. **(B)** Relative expression levels of genes enriched in dorsal and ventral CA3 in control, COUP-TFI cKO, and cTG Hp. **(C)** Immunostaining of protein kinase c delta (Prkcd, green) and calretinin (Calb2, red) on P7 coronal sections of COUP-TFI cKO, control, and COUP-TFI cTG Hp at rostral and caudal levels. (c₁-c₄) In controls, the expression of Prkcd and Calb2 was restricted to rostral and caudal CA3, respectively. Ectopic expression of Calb2 (arrowheads) was detected in rostral CA3 in COUP-TFI cTG. **(D)** Diagrams demonstrating the locations of dorsal and ventral CA1, CA2, and CA3 in COUP-TFI cKO, control, and cTG Hp, based on the expression of subregion-specific marker genes. The extent of each marker gene expression domain was measured by staining serial coronal sections. The position and relative length of each domain to the total length of CAs (from the distal end of CA1, top, to the proximal end of CA3, bottom) is plotted along the rostrocaudal axis (rostral to the left and caudal to the right). An expansion of dorsal domains was found in the cKO, while ectopic ventral domains were present in the cTG. Scale bars, 1 mm (A, C), 100 μm (a₁-a₄, c₁-c₄). DG, dentate gyrus; Sub, subiculum. Statistical analyses were performed using Student *t* test (B). The data underlying this figure can be found in S1 Data.

(COUP-TFI^TG/O; Nex-Cre) mice, we did not detect changes in CA1 thickness, lamination, or the expression patterns of Wfs1 and Dcn (S10 Fig). This lack of effect suggested that COUP-TFI functions in progenitors and not in postmitotic neurons.

We next performed chromatin immunoprecipitation (ChIP)-seq for COUP-TFI to identify its downstream targets in E13.5 cortices. The candidate COUP-TFI downstream genes were enriched in several key signaling pathways, including VEGF,

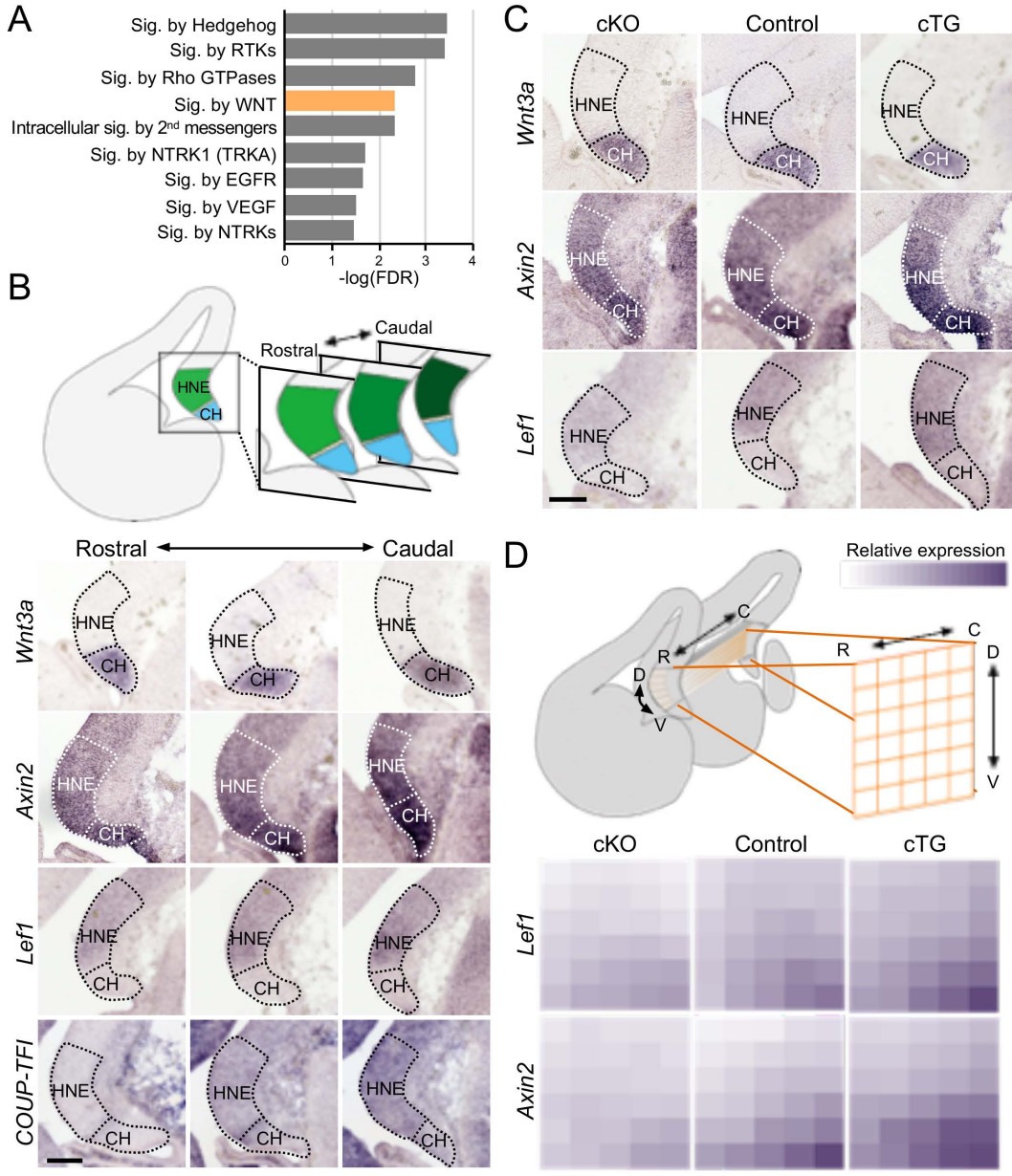

**Fig 4. COUP-TFI modulates the expression of Wnt-downstream genes in the embryonic hippocampal neuroepithelium. (A)** GO pathway enrichment of COUP-TFI downstream genes in embryonic cortex (E13.5). **(B)** In situ hybridization of *Wnt3a*, Wnt-downstream genes, *Axin2* and *Lef1*, and *COUP-TFI*, in the medial pallium along the rostral to caudal axis at E13.5. *Wnt3a* was expressed in the cortical hem (CH). *Axin2*, *Lef1*, and *COUP-TFI* all showed high-caudal-to-low-rostral expression gradients in the hippocampal neuroepithelium (HNE). **(C)** At E13.5, the expression levels of *Lef1* and *Axin2* were down-regulated in COUP-TFI cKO and up-regulated in COUP-TFI cTG in HNE. Wnt3a expression in CH of COUP-TFI mutants was similar to controls. **(D)** Quantitative analyses of *Axin2* and *Lef1* RNA expression levels in HNE. Both *Axin2* and *Lef1* RNA showed a high-caudal-to-low-rostral gradient in controls. The gradient was diminished in COUP-TFI cKO and enhanced in COUP-TFI cTG HNE. C, caudal; D, dorsal, R, rostral; V, ventral. Scale bars, 100 μm.

EGFR, Wnt, and hedgehog pathways (Fig 4A). As the Wnt signaling pathway was previously shown to be critical for Hp development [18,33], we examined the expression of genes that are both involved in Wnt signaling and have COUP-TFI binding sites in their regulatory regions (e.g., *Axin2* and *Lef1*) in the medial pallium. While the expression level of Wnt ligand *Wnt3a* gene was unchanged in the CH across the rostrocaudal axis, the expression patterns of *Axin2* and *Lef1* each showed a high-caudal-to-low-rostral gradient in the HNE (the Hp primordium), correlated with the expression gradient of COUP-TFI (Fig 4B). This suggested that, in addition to being regulated by Wnt signaling, *Axin2*, and *Lef1* expression are likely to be regulated by COUP-TFI.

We further compared the expression levels of *Wnt3a*, *Axin2*, and *Lef1* in the medial pallium of control and COUP-TFI mutant cortices. While *Wnt3a* expression remained unchanged, both *Axin2* and *Lef1* in the HNE were down-regulated in the cKO and up-regulated in the cTG (Fig 4C). Quantifying the intensity of the in situ hybridization signals of *Axin2* and *Lef1* in the HNE on serial sections along the rostrocaudal axis, we found that both *Axin2* and *Lef1* expression was dramatically attenuated in the cKO and up-regulated and rostrally shifted in the cTG (Fig 4D). Such shifts in the expression gradient of the Wnt signaling downstream genes in the medial pallium correlate with the patterning phenotypes observed in the COUP-TFI mutant Hp. Our findings suggested that COUP-TFI regulates Hp patterning through an interplay with Wnt signaling coming from the CH.

## Discussion

Although much is known about hippocampal structures, functions, and neuronal properties, the genetic programs governing Hp development remain largely unclear. In this study, we identified a key mechanism for patterning Hp subregions, demonstrating that the high-caudal-to-low-rostral expression gradient of COUP-TFI in Hp progenitors directs the fate of Hp subregions. Specifically, higher levels of COUP-TFI expression promote the vHp identity. This conclusion was made based on our comprehensive analyses of Hp cytoarchitecture and gene expression patterns across the entire Hp in the COUP-TFI mutants. This whole-Hp analysis approach enabled us to capture phenotypes that might have been missed by examining marker gene expression on individual sections. In the COUP-TFI cKO, we observed expanded expression domains of genes typically enriched in the dorsal CA1 and CA3 and a corresponding restriction of expression domains for genes normally enriched in ventral CA1 and CA3. Conversely, in COUP-TFI cTG, the overexpression of COUP-TFI resulted in the ectopic formation of ventral CA1 and CA3 regions in the dorsal hippocampus, replacing parts of the dorsal CA1 and CA3 (Fig 3D). In conclusion, our study identified COUP-TFI as an intrinsic regulator of Hp subregion patterning, with its gradient of expression playing a pivotal role in distinguishing dorsal from ventral hippocampal identities.

### Determination and segregation of subregions in the hippocampus

Previous electrophysiological, anatomical, and molecular studies suggested that Hp is organized in a graded fashion along the dorsal-ventral axis [34,35]. For example, the size of place fields gradually increases from dorsal to ventral in the Hp, indicating a graded functional organization [36]. Furthermore, Hp innervation with the amygdala and the anterior olfactory nucleus follows a topographical gradient, wherein the pyramidal neurons located more ventrally project to more medial portions of these subcortical structures [37, 38]. Gene expression gradients are also present in the Hp along the dorsoventral axis. Additionally, different sets of genes have been shown to be enriched in specific subregions of the adult Hp, including the dHp, vHp, and the intermediate Hp, although the precise boundaries of these subregions have not been clearly defined [13,39].

In this study, we showed that even at an early postnatal stage, when not all the subregion-specific genes are fully expressed, the Hp can already be divided into dorsal and ventral compartments using a handful of marker genes. Our findings align with previous work showing that within the first postnatal week in rats, the connectivity between Hp and entorhinal cortex is established in an adult-like topography and several dorsally or ventrally enriched genes are expressed at the respective pole of the neonatal Hp [40]. This suggests that Hp compartmentalization is initiated early

 

in development, well before the onset of mature hippocampal functions at 3–4 weeks postnatally in mice [41,42]. Based on the early intrinsic genetic patterning, the functions of Hp subregions are likely to be further refined by extrinsic factors, such as sensory input, which has been shown to be important for hippocampal maturation [43].

Interestingly, when COUP-TFI expression was elevated in cTG mice, we observed the formation of ectopic vCA1 and CA2 domains within the dCA1, as well as ectopic vCA3 within the dCA3. This finding is consistent with our previous study, in which COUP-TFI overexpression led to the emergence of ectopic medial entorhinal cortex (MEC) domains in the neocortex (NC) [24]. In that study, we showed that cortical progenitors were multipotent and could give rise to either MEC or NC neurons. The fate choice of these progenitors was determined by two patterning TFs with opposing expression gradients: high level of COUP-TFI (expressed in a high-caudal-to-low-rostral expression gradient in cortical progenitors) promoted MEC cell fate, while another, yet undefined TF (expressed in a high-rostral-to-low-caudal gradient) likely promoted NC cell fate. In our cTG model, COUP-TFI overexpression created a condition in which both patterning TFs were expressed at high levels, resulting in a zone of fate instability where progenitors could adopt either fate. Subsequently, the physical segregation of these distinct cell populations was driven by differential cell affinities.

In the current study, we propose that a similar mechanism governs the Hp dorsal-ventral patterning. COUP-TFI, which is normally expressed in a high-caudal-to-low-rostral gradient in HNE, appears to promote ventral hippocampal identity. Its ectopic overexpression in the rostral HNE, which normally gives rise to dorsal Hp, generates an instability zone where both dorsal and ventral Hp neuronal types are produced. These dHp and vHP neurons then undergo segregation, likely via differential cell adhesion, leading to the formation of ectopic ventral domains within dorsal hippocampal regions. Overall, our findings suggest that distinct hippocampal subregions, including dorsal and ventral CA1 and CA3, are likely segregated by differential cell adhesion affinities, and COUP-TFI plays a key role in patterning of these subregions.

Due to the lack of specific markers for dorsal and ventral CA2, we could not determine whether the dorsoventral patterning changes could also be found in CA2 in the COUP-TFI mutant Hp. However, the observation of ectopic Pcp4+ clusters in the CA1 of COUP-TFI cTG suggests that similar adhesion-based mechanisms may also underlie the segregation of CA1 and CA2 neurons.

## COUP-TFI regulates the Wnt signaling pathway during Hp development

COUP-TFI is expressed in a high-caudal-to-low-rostral gradient among Hp progenitors, and this expression pattern correlates with its function; a high level of COUP-TFI promotes the caudally located vHp. The role of COUP-TFI in patterning the Hp is similar to its role in the area patterning of the neocortex [44,45], suggesting that similar mechanisms may operate across these regions. As Hp is part of the archicortex, a phylogenetically older region of the cerebral cortex, this TF gradient-based patterning mechanism may represent an evolutionarily conserved process.

COUP-TFI was previously implicated in several aspects of cortical development, including neurogenesis and neuronal specification, through its regulation of multiple genetic pathways [45–47]. Here, we identified Wnt signaling pathway as one of the COUP-TFI downstream genetic networks involved in Hp patterning. Wnt signaling was shown to regulate Hp development. For example, Wnt3a expressed in the CH is required for Hp outgrowth [33,48]. Our results showed that COUP-TFI regulates several Wnt downstream genes, including *Lef1* and *Axin2*, to be expressed in the medial pallium in a high-caudal-to-low-rostral gradient. This expression gradient of Wnt downstream genes implies that Wnt signaling is likely to be involved in Hp patterning. Although our study showed that COUP-TFI could positively regulate the expression gradient of Wnt downstream genes along the rostral-caudal axis in the hippocampal progenitors, previous work demonstrated that COUP-TFI represses β-catenin signaling in cortical progenitors [45]. This difference in findings suggests that the functions of COUP-TFI in regulating the Wnt signaling pathway might be context-dependent, with different effects in distinct cortical regions.

Given that different Hp subregions are associated with distinct functions, we anticipate that altering the relative size of these subregions would have behavioral consequences. However, we did not explore this aspect with our current animal models because COUP-TFI expression was altered across the entire cerebral cortex in them. Changes in COUP-TFI

expression would likely affect other cortical regions, such as entorhinal cortex, which is involved in spatial navigation, and amygdala, which plays a role in emotion processing. The broad effects make these animals unsuitable for isolating the behavioral outcomes of hippocampal mispatterning. Nevertheless, loss-of-function mutations of the *COUP-TFI* gene are known to cause Bosch-Boonstra-Schaaf optic atrophy syndrome in humans, a rare developmental disorder [49,50] in which patients often display intellectual disabilities [46]. This highlights the significance of COUP-TFI in cognitive development.

The dHp and vHp differ not only in function and molecular properties but also in disease susceptibility. For example, the dHp CA1 exhibits greater vulnerability to ischemic insults [51,52], while the vHp is closely linked to conditions such as depression [53], temporal lobe epilepsy [54,55], schizophrenia [56], and Alzheimer's disease [57,58]. Understanding the mechanisms of Hp patterning, such as those involving COUP-TFI, may provide valuable insights into the links between Hp development, function, and disease vulnerability. Further studies are essential to unravel the full implications of hippocampal patterning in health and disease.

## Materials and methods

### Ethics statement

All experimental procedures involving mice were conducted in accordance with the approved animal protocol (AS IACUC Protocol ID: 24-07-2198) by the Institutional Animal Care and Utilization Committee at Academia Sinica. The procedures strictly follow the guidelines for the care and use of laboratory animals established by the Council of Agriculture in Taiwan.

### Animals

COUP-TFI floxed and transgenic mice were provided by M.-J. Tsai, and Emx1-Cre mice were obtained from K. Jones. Controls and conditional COUP-TFI cKO and cTG mice were generated by mating COUP-TFI$^{f/f}$ or COUP-TFI$^{f/+}$ with COUP-TFI$^{f/+}$; Emx1$^{Cre/+}$ and COUP-TFI$^{TG/TG}$ with Emx1$^{Cre/+}$ mice. The mice were housed in a climate-controlled room with a 12h light/dark cycle, and provided food and water ad libitum. The identification of a vaginal plug and the day of birth were designated as E0.5 and P0, respectively. All control and COUP-TFI mutant mice were littermates, with mouse genotypes determined by PCR as previously described [24].

### Tissue clearing and light-sheet microscopic image reconstruction

Fixed brains from postnatal day 60 (P60) male reporter mice were processed using a modified PEGASOS-based tissue clearing protocol. Samples were first decolorized in a solution containing 25% N,N,N′,N′-Tetrakis(2-hydroxypropyl) ethylenediamine (Quadrol) and 5% ammonium hydroxide for 2 d, with daily solution changes. Delipidation was carried out in a graded series of tert-butanol (tB) solutions with 3% Quadrol: 30% tB for 4 h, 50% tB for 6 h, and 70% tB for 24 h. This was followed by dehydration in 75% tB and 3% Quadrol for 24 h. Final optical clearing was achieved in a PEG-based medium consisting of 75% benzyl benzoate (BB), 22% polyethylene glycol methacrylate (PEGMMA), and 3% Quadrol for another 24 h. Imaging was performed using a custom-built scanning Bessel beam light-sheet fluorescence microscope controlled by Micro-Manager software. The system employed dual-sided illumination from two opposing 4× dry objectives (Olympus, NA 0.28), arranged symmetrically on either side of the sample chamber and directed through flat windows. A matched 4× detection objective (Olympus, NA 0.28), fitted with a custom-designed protective cap, was positioned orthogonally to the illumination axis to allow immersion into the clearing medium. The system was equipped with a four-line laser light source, and fluorescence signals were captured using a scientific CMOS camera (Andor Zyla 4.2 Plus). Volumetric datasets were acquired as sequential x–y plane slices. 3D reconstructions were generated using Maximum Intensity Projection (MIP) and volume rendering algorithms. YFP-positive (YFP$^+$) neurons and hippocampal boundaries were segmented and analyzed using Imaris software (Oxford Instruments).

## Tissue preparation

To characterize the histological features of the Hp, control, COUP-TFI cKO and cTG mice at P7 or P60 were anesthetized and perfused transcardially with 4% formaldehyde in phosphate-buffered saline (PBS). The brains were then isolated and further fixed in 4% formaldehyde at 4 °C overnight, followed by dehydration in 30% sucrose solutions. The flattened cortices were fixed in 4% formaldehyde between two glass slides at 4 °C overnight. For experiments on embryonic stage animals, embryonic day 13.5 (E13.5) embryos were collected from deeply anesthetized pregnant mice. The embryonic heads were immediately removed for fixation at 4 °C overnight followed by dehydration.

## Nissl, immunofluorescence, and EdU staining

For frozen tissue sectioning, brains were embedded in Tissue-Tek optimal cutting temperature (OCT) compound (Sakura Fientek), and sectioned using a cryostat (Leica). Coronal sections were prepared at 25 µm thickness for P7 and P60 mice. For Nissl staining, brain sections were rinsed with water and stained for 10 min in a solution containing 0.2% cresyl violet and 0.5% acetic acid. Following a brief rinse with water, the sections were dehydrated through an ascending alcohol series (70%, 95%, and 100%) for 1 min each, then cleared in xylene and mounted in neutral mounting medium DPX (Sigma). For immunostaining, tissue sections were subjected to antigen retrieval in boiling 10 mM sodium citrate buffer (pH 9.0) for 10−20 min, followed by cooling for 30 min. Permeabilization and blocking were performed with phosphate-buffered saline (PBS) containing 0.3% Triton X-100 and 3% bovine serum albumin (BSA) for 1 h at room temperature. Sections were then incubated overnight at 4 °C with primary antibodies at the specified concentrations (detailed in S1 Table). Following three washes with PBS, sections were incubated with corresponding fluorescence-conjugated secondary antibodies for 1 h at room temperature. For EdU (5-ethynyl-2′-deoxyuridine) labeling, EdU (500 ng) was injected into timed-pregnant mice, and the EdU-positive cells were detected with a Click-iT EdU imaging kit (Invitrogen). Nuclei were counterstained with 4′,6-diamidino-2-phenylindole (DAPI).

## RNA extraction and real-time polymerase chain reaction (PCR)

Total RNA was extracted from hippocampal tissues using Trizol reagent (Invitrogen) and reverse transcribed with an oligo-dT primer and reverse transcriptase (Promega). Quantitative PCR (qPCR) was performed using the asymmetrical cyanine dye SYBR Green I mixture (Roche) and a LightCycler 480 system. Data analysis was conducted according to the comparative Ct (threshold cycle) method. Primers used are listed in S2 Table.

## In Situ hybridization

In situ hybridization was performed as previously described [24]. Antisense RNA probes (primers for generating transcription templates listed in S3 Table) were labeled with digoxigenin (DIG) using a DIG-RNA labeling kit (Roche). Coronal sections, prepared at 25 µm thickness for P7 and at 20 µm for E13.5 embryos, were pretreated with 10 µg/mL proteinase K at room temperature for 10 min, followed by prehybridization for 1 h. Sections were then hybridized with RNA probes at 65 °C overnight. After posthybridization washes, the sections were incubated with alkaline phosphatase-conjugated anti-DIG antibody (Roche) overnight at 4 °C. The distribution of RNA probes was visualized by chromogenic staining with 0.34 mg/mL NBT (nitro blue tetrazolium chloride) and 0.18 mg/mL BCIP (5-bromo-4-chloro-3-indolyl phosphate).

## Morphometric analysis, image acquisition, and quantification

For morphometric analysis of the hippocampus and its subregions, a series of coronal sections was prepared at a thickness of 25 µm. Every sixth section along the rostrocaudal axis was stained for Nissl or specific neuronal markers. Images were captured using an upright microscope (Axio Imager A2, Zeiss) equipped with an automated system or a high-throughput slide scanner (TissueGnostics). Cell density, laminar thickness, and area were quantified using ImageJ.

To estimate total volumes and cell numbers, values obtained from the sampled sections were multiplied by a factor of six (every sixth section was examined). To ensure consistent comparison of patterning changes across samples despite hippocampal size variability, section positions were scaled to a proportional axis. To determine gene expression gradient (Fig 4D), HNE was divided into six equally spaced domains on five adjacent coronal sections. The average intensity of RNA in situ hybridization color reaction signals was measured within each domain. These intensity values were normalized to the background level of each section. Relative expression levels were then calculated by comparing the average intensity of each domain to the highest expression observed, which was in the most caudal ventral domain. The average intensity values of each domain were then visualized as a heatmap to illustrate the gradient.

### Chromatin immunoprecipitation sequencing (ChIP-seq)

The dorsal telencephalon was isolated from E13.5 wild-type mouse embryos. The tissues were dissociated and crosslinked with 2 mM Di(N-succinimidyl) glutarate (DSG), before fixation in 1% formaldehyde. The crosslinking reaction was quenched with 125 mM glycine. The cell lysates were sonicated to generate chromatin fragments averaging 100–300 bp in length. Chromatin-protein complexes were then immunoprecipitated overnight at 4 °C using either mouse anti-COUP-TFI antibody or mouse gamma globulin (Jackson ImmunoResearch) as a control. The COUP-TFI-bound chromatin complexes were captured with Dynabeads Protein G (Invitrogen) at 4 °C for 2 h. Final chromatin fragments were extracted and sequenced using the Illumina NextSeq platform. Reactome enrichment analysis was performed using a PANTHER website [59].

### Data presentation and statistical analysis

All data are expressed as mean ± SEM and were analyzed using SigmaPlot software. Statistical significance was determined using one-tailed Student $t$ test for comparisons between genotypes with Bonferroni correction applied due to repeated comparisons. For comparisons along the rostrocaudal axis, the Mann–Whitney $U$ test with Bonferroni correction was used. Significance in figures is denoted as follows: $*P < 0.05$; $**P < 0.01$; $***P < 0.001$ for comparisons between genotypes, and $^{\#}P < 0.05$; $^{\#\#}P < 0.01$; $^{\#\#\#}P < 0.001$ for comparisons between rostral and caudal sections. For Mann–Whitney results with correction, thresholds were adjusted to $*P < 0.025$; $**P < 0.0025$; $***P < 0.001$, respectively.

### Supporting information

**S1 Fig. Consistent cortical surface area between control and COUP-TFI mutant mice (Related to Fig 1). (A)** Dorsal views of the brain from control and COUP-TFI mutant mice at P7. **(B)** Comparisons of brain surface area, length, and width in control and COUP-TFI mutant mice revealed no significant differences (N.S.). Scale bar, 5 mm. Statistical analyses were performed using Student $t$ test. The data underlying this figure can be found in S1 Data.
(S1_Fig.TIFF)

**S2 Fig. Rostrocaudal changes of the Hp in COUP-TFI mutant mice (Related to Fig 1). (A)** Serial Nissl-stained coronal sections of P7 control, COUP-TFI cTG, and cKO hippocampus along the rostrocaudal axis. **(B, C)** Distribution of EdU-labeled cells derived at E13.5 and E15.5 in rostral and caudal CA1 (B) and CA3 (C) PCL of COUP-TFI cKO, control and COUP-TFI cTG Hp at P0. **(D, E)** Quantification of EdU-labeled cell number in the deep and superficial sublayers of CA1 (D) and CA3 (E) regions at the rostral and caudal levels in COUP-TFI cKO, control and COUP-TFI cTG Hp at P0. Scale bars, 1 mm (A); 50 μm (B, C). Statistical analyses were performed using Student $t$ test. The data underlying this figure can be found in S1 Data.
(S2_Fig.TIFF)

**S3 Fig. Changes in CA1 thickness and layer organization in COUP-TFI mutant Hp (Related to Fig 2). (A, B)** Quantification of layer thickness (A) and cell densities (B) in stratum oriens (SO) and stratum radiatum (SR) in COUP-TFI cKO, control, and COUP-TFI cTG Hp at rostral and caudal levels. **(C, C')** Morphometric analysis of PCL thickness between COUP-TFI cKO, control and COUP-TFI cTG mice along the rostrocaudal axis. Position number is based on Fig 1D, with normalized positions indicated on the right (C'). **(D)** Coronal sections of rostral and caudal Hp of P7 COUP-TFI cKO, control and COUP-TFI cTG mice stained with Ctip2 (green) and Satb2 (red). Enlarged images of insets are shown on the right ($d_1$-$d_6$). **(E)** The distribution of Ctip2$^-$ Satb2$^+$ (red), Ctip2$^+$ Satb2$^-$ (green), and Ctip2$^+$ Satb2$^+$ (yellow) neuronal populations across the CA1 pyramidal cell layer in COUP-TFI cKO, control, and COUP-TFI cTG hippocampus at the rostral and caudal levels. Scale bars, 1 mm (D); 100 μm ($d_1$-$d_6$). Statistical analyses were performed using Student $t$ test (A, B) and Mann–Whitney $U$ test (C, C'). The data underlying this figure can be found in S1 Data.
(S3_Fig.TIFF)

**S4 Fig. Complementary distribution of Wfs1$^+$ and Dcn$^+$ neuron populations in CA1 along the rostrocaudal axis (Related to Fig 2).** Serial images of immunostaining for Wfs1 (green) and Dcn (red) in coronal sections of P7 control and COUP-TFI cTG CA1, and quantification of signal intensities along the rostral to caudal axis. Neurons in the rostral CA1 were mostly Wfs1$^+$ and those in the caudal CA1 were mostly Dcn$^+$ in control. In COUP-TFI cTG, patches of Wfs1$^+$ and Wfs1$^-$ neurons were found in rostral CA1. Dcn$^+$ neurons were ectopically identified in the Wfs1$^-$ domain, adjacent to Wfs1$^+$ neurons in rostral CA1 in COUP-TFI cTG. However, the ectopic Dcn$^+$ cells were not present in most examined sections. Scale bars, 100 μm.
(S4_Fig.TIF)

**S5 Fig. Dorsal-ventral patterning changes in COUP-TFI mutant CA1 observed in flattened hippocampus (Related to Fig 2). (A)** Diagram illustrating the flattening and sectioning. P7 cortical hemispheres were dissected and flattened between two glass slides ($a_1$). Transverse hippocampal sections were obtained by slicing perpendicularly to a line connecting the dorsal and ventral poles ($a_2$). **(B)** Nissl staining (left) and immunostaining for Wsf1 (green, a dorsal CA1 marker) and Dcn (red, a ventral CA1 marker) in transverse sections of the hippocampus from indicated genotypes. **B)** Nissl staining (left) and immunostaining for Wsf1 (green, a dorsal CA1 marker) and Dcn (red, a ventral CA1 marker) in transverse sections of the hippocampus from indicated genotypes. The size of Wfs1 expression domains (between arrowheads) was increased in cKO and decreased in cTG. Ectopic Dcn expression was found in the dorsal hippocampus of cTG (*). Scale bar, 500 μm.
(S5_Fig.TIFF)

**S6 Fig. Identification of genes enriched in Hp subregions (Related to Figs 2 and 3). (A)** UMAP representation of excitatory neurons from hippocampal CA regions (based on scRNAseq database [30]) colored according to CA region. **(B)** UMAP representations of dorsal and ventral CA1 and CA3 subregions, as well as CA2. **(C)** Representative genes enriched in the dorsal and ventral CA1 and CA3 regions. The data underlying this figure can be found in S1 Data.
(S6_Fig.TIFF)

**S7 Fig. Changes in the expression of genes enriched in vCA1 in COUP-TFI mutant CA1 (Related to Fig 2).** In situ hybridization of *Dcn*, *Cpen7*, and *Nnat* on rostral and caudal coronal sections of P7 control and COUP-TFI mutant Hp. Within CA1, these genes are enriched in the vCA1. The expression of these markers was ventrally shifted in the cKO (arrowheads). Ectopic expression domains were identified in the cTG (marked by asterisks). Scale bar, 500 μm.
(S7_Fig.TIFF)

**S8 Fig. Expression of CA2 markers, Pcp4 and Rgs14, in COUP-TFI mutants (Related to Fig 3). (A)** Immunostaining and quantification of signal intensities for Pcp4 (green) and Satb2 (red) in P7 serial coronal sections of control and COUP-TFI cTG hippocampi along the rostral to caudal axis. **(B)** Immunostaining of Pcp4 and Rgs14 in P7 serial coronal sections

of COUP-TFI cKO, control, and COUP-TFI cTG Hp. In the ectopic Pcp4 expression domains within the COUP-TFI cTG CA1 PCL (solid arrowheads), higher Rgs14 expression was detected. However, Rgs14 was not expressed in the Pcp4$^+$ cells in the deep layer of CA1 in the caudal CA1 (open arrowheads). Scale bars, 1 mm (B), 100 μm (A and b$_1$-b$_4$). DG, dentate gyrus; i., intensity; Sub, subiculum.
(S8_Fig.TIFF)

**S9 Fig. Serial sections of markers associated with altered dorsoventral patterning in the hippocampi of COUP-TFI mutants (Related to Figs 2 and 3).** Serial images from immunostaining revealed changes in distinct markers, including Wfs1 (green) and Dcn (red) **(A),** Prkcd (green) and Calb2 (red) **(B)**, and Pcp4 (green) and Satb2 (red) **(C)**, along the rostrocaudal axis in hippocampi from control and COUP-TFI mutants. Scale bars, 1 mm.
(S9_Fig.TIF)

**S10 Fig. COUP-TFI overexpression in postmitotic neurons does not affect Hp patterning (Related to Fig 2). (A)** Similar hippocampal structures were detected in Nissl-stained P7 coronal sections of control and COUP-TFI cTG-Nex (COUP-TFI$^{TG/O}$; Nex$^{Cre/+}$) cortices at rostral and caudal levels. Enlarged images of insets are shown on the right (a$_1$-a$_4$). **(B, C)** Immunostaining of Ctip2 (green) and Satb2 (red) (B) and Wfs1 (green) and Dcn (red) (C) in rostral and caudal hippocampi of control and COUP-TFI cTG-Nex revealed similar lamination in both rostral and caudal CA1. Enlarged images of insets are shown on the right (b$_1$-b$_4$, c$_1$-c$_4$). DG, dentate gyrus; Sub, subiculum. Scale bars, 1 mm (A-C), 100 μm (a$_1$-c$_4$).
(S10_Fig.TIFF)

**S1 Movie. 3D reconstruction of light sheet microscopic images from cleared brains of Thy1-YFP transgenic mice (Related to Fig 1).** The density of YFP-expressing neurons varied along the rostrocaudal axis of the hippocampus (Hp). Compared with control mice, the Hp was enlarged in the cTG. In contrast, YFP expression in the Hp was reduced in the cKO, whereas the domain of strong YFP signals was expanded in the cTG.
(S1_Movie.MOV)

**S1 Table. Antibody list.**
(S1_Table.PDF)

**S2 Table. PCR primer list.**
(S2_Table.PDF)

**S3 Table. ISH primer list.**
(S3_Table.PDF)

**S1 Data. Source data for main and supporting figures.**
(S1_Data.XLSX)

## Acknowledgments

We thank Dr. Ming-Jer Tsai for providing the *COUP-TFI* floxed and transgenic allele, Dr. Kevin Jones for *Emx1*-Cre, and Dr. Chau-Ren Jung for providing statistical consultation. We also thank Dr. Wen-Hsin Hsu and other members of the Chou laboratory for their help.

## Author contributions

**Conceptualization:** Ching-San Tseng, Shen-Ju Chou.

**Data curation:** Ching-San Tseng.

**Formal analysis:** Ching-San Tseng, Zi-hui Zhuang, Hsiang-Wei Hsing, Yu-Kuan Pan.

**Funding acquisition:** Shen-Ju Chou.

**Investigation:** Ching-San Tseng, Zi-hui Zhuang, Hsiang-Wei Hsing, Shen-Ju Chou.

**Methodology:** Chia-Ming Lee, Tsan-Ting Hsu, Yi-Ping Hsueh, Bi-Chang Chen.

**Resources:** Chia-Ming Lee, Tsan-Ting Hsu, Yi-Ping Hsueh, Bi-Chang Chen.

**Supervision:** Shen-Ju Chou.

**Validation:** Ching-San Tseng.

**Visualization:** Ching-San Tseng, Zi-hui Zhuang.

**Writing – original draft:** Ching-San Tseng, Shen-Ju Chou.

**Writing – review & editing:** Shen-Ju Chou.

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
