## [Editor Report · Decision Letter 0]

3 Feb 2025

Dear Dr Chou, 

Thank you for submitting your manuscript entitled "Determining the relative sizes of hippocampal subregions along the dorsal-ventral axis by COUP-TFI expression gradient" for consideration as a Short Reports by PLOS Biology.

Your manuscript has now been evaluated by the PLOS Biology editorial staff, as well as by an academic editor with relevant expertise, and I am writing to let you know that we would like to send your submission out for external peer review.

Once your full submission is complete, your paper will undergo a series of checks in preparation for peer review. After your manuscript has passed the checks it will be sent out for review. To provide the metadata for your submission, please Login to Editorial Manager (https://www.editorialmanager.com/pbiology) within two working days, i.e. by Feb 05 2025 11:59PM.

Kind regards,

Taylor

Taylor Hart, PhD, 

Associate Editor

PLOS Biology

thart@plos.org

---

## [Decision Letter · Decision Letter 1]

12 Mar 2025

Dear Dr Chou,

Thank you for your patience while your manuscript "Determining the relative sizes of hippocampal subregions along the dorsal-ventral axis by COUP-TFI expression gradient" was peer-reviewed at PLOS Biology. It has now been evaluated by the PLOS Biology editors, an Academic Editor with relevant expertise, and by several independent reviewers. 

In light of the reviews, which you will find at the end of this email, we would like to invite you to revise the work to thoroughly address the reviewers' reports.

Reviewers called the paper descriptive but interesting and potentially important, and R2 is very positive and only raises minor points. However, the other reviewers noted several areas where the conclusions require further support. R1 pointed out some under-explored areas in the paper and missing technical details. R3 and R4 raise several concerns about the alignment and delineation of hippocampus regions, based on anatomy and molecular markers. These reviewers also pointed out areas where discrepancies with prior literature were not well addressed.

We would like to invite a Major Revision of this submission. In line with our discussion with the Academic Editor, we must emphasize the need to perform new experiments and analyses to better delineate the differences in hippocampus subregions between mouse genotypes, including through adding data from additional molecular markers. In particular, you should carefully consider the points raised by R4 regarding hippocampus organization and thoroughly address them in your response. You should also revise the text to better integrate the findings with prior literature and include missing technical details, and otherwise address the reviewers' concerns. 

Given the extent of revision needed, we cannot make a decision about publication until we have seen the revised manuscript and your response to the reviewers' comments. Your revised manuscript is likely to be sent for further evaluation by all or a subset of the reviewers.

**IMPORTANT - SUBMITTING YOUR REVISION**

*Re-submission Checklist*

*Published Peer Review*

*PLOS Data Policy*

*Blot and Gel Data Policy*

Sincerely,

Taylor

Taylor Hart, PhD, 

Associate Editor

PLOS Biology

thart@plos.org

REVIEWS:

Reviewer #1: Tseng et al. work focuses on understanding the role of COUP-TF1, a transcription factors, in the development of the hippocampus, particularly, in the patterning of hippocampal subregions. The authors use a combination of anatomical and molecular approaches to analyze how a deficit (cKO) or an overexpression (cTG) of COUP-TF1 affects the sizes of the different subregions. 

The work is mainly descriptive (which I don't see as a negative point), and the authors have carefully analyzed the changes in the cKO and cTG with different markers and across the whole dorso-ventral axes. I particularly appreciate having supplementary figures with additional images to show stainings at different rostro-caudal levels. I feel few points could be better investigated or discussed, but I overall think this work will be rather interesting for the field of developmental neuroscience and those studying the hippocampal regions. 

Major comments:

* The authors show how changes in the level of COUP-TF1 often leads to the expression of region-specific markers in ectopic regions. For example, the expression of Dcn in patches of CA1 or of Pcp4 in deep PCL in caudal CA1. This is very interesting, though it is not very much discussed in the manuscript. What could be the reason of these ectopic expression? In particular, in the Discussion, the authors mention that their data support the notion that CA1 PC and CA3 PC derive from different progenitors, but vCA1 and dCA1 would derive from the same progenitors. If that is the case, how do we explain the ectopic cells? Are those the consequence of migration defects (like abnormal cells migrating in the wrong place). Or are they the product of misspecification of specific cells from some specific progenitors? I would appreciate if the authors could speculate in the discussion, as these data might be difficult to interpret for scientists outside of the field.

* The authors have some data with Edu labeling in the supplementary figures, where they show that there is no increase in cell production. These data nicely fit with what is reported in Fig.2B. However, I am sure that the authors known that the literature has showed a difference in the DOB of deep VS superficial and that CA3 and CA1 cells follow different timeline for DOB and peak of neurogenesis. It would be extremely interesting if the authors could add i) analyses of the deep VS superficial part of the PCL (a simple 50-50 division could work, or authors could use specific markers to distinguish them); and ii) also similarly analyses of CA3. This would help in establishing if the differences observed are also caused by a mis-timed differentiation or neurogenesis.

* Fig. 4B. The high-caudal-low-rostral gradients are a bit hard to catch, as they seem a bit subtle. In particular, it almost seems like it is more the size of the HNE that changes, rather than the signal intensity (in particular for Axin2). Perhaps the authors could use some dashed lines to show the areas we should focus on and have some higher magnification images that better support their claim. 

Minor comments: 

* From Fig.2A-2c, it is unclear what is counted in the B and C. The text seems to suggest all the CA1 area, but the figure and legend say "CA1 PCL". The authors should specify. And since it seems that also the neuropil is affected in the cKO and the cTG, the authors might consider analyzing those differences and discuss what could be the reason, in case they have analyzed only the PCL.

* It is rather curious that the authors never analyze the Dentate Gyrus. Since the paper is about subregions of the hippocampus, there could be an expectation that this subregion is also included in the analysis. While I am not asking the authors to add work on the DG, perhaps they should specify in the text (and perhaps the title) that they are analyzing only the CA regions, which could be called "Hippocampus proper". 

* For the experiments in Fig.S2, do authors count only cells in the PCL? If so, they should specify it. 

* Related to the Edu experiments, they seem to be completely missing from the Methods section. Authors should add that. 

* Fig.S4, legend: typo in Reference for RNA-seq database (I guess it's #30 and not #3).

* Fig.4D, authors should explain somewhere how the quantification was done and how the "square" was obtained. 

* The methods section in general is lacking a bit in details regarding equipment models and parameters used. For example, nothing about that is reported for the Light-Sheet microscope. It would be nice if the authors could add that and re-check their methods section. 

* Fig.2. In D, an explanation of the arrow heads is missing. In G, I assume V/D cell ratio means the authors first did the Dcn/Wfs1 ratio for the Ventral and Dorsal region and then calculated that ratio. That could be explained a bit better in the legend.

* I couldn't find the Supplementary Movie 1 in my files. 

-----------

Reviewer #2: Outstanding and novel results on the role of Nr2f1 in regulating regionalization of the mouse hippocampus studied using Nr2f1 loss and gain of function mutants. The figures are excellent. Quantifications are carefully done. The writing is precise.

Therefore, the paper Merits publication with some minor editing and no further review.

For the Intro and/or Discussion, please reference Faedo et al, Cerebral Cortex, 2007 and Ypsilanti et al PNAS 2021. These papers showed that in the cerebral cortex, Nr2f1 alone or with Nr2f2, promote ventral identity (see Figure 3 of PNAS paper). Figure 8 of cerebral cortex paper showed that loss and gain of Nr2f1 regulates D/V patterning of the cortex. While it is well known that Nr2f1 promotes caudal identity, it is not well appreciated that it also has a key role in patterning the ventral cortex. This is highly relevant to Nr2f1's role in hippocampal D/V patterning.

Regarding the NR2F1 ChiP-seq - please reference Ypsilanti et al PNAS 2021 who also performed this type of analysis on the E12.5 cortex, although they focused on the neocortex and Nr2f1 binding to regulatory elements of other TFs. They did report that Nr2f1, Pax6, Emx2 Lhx2 and Pbx1 showed co-binding to regulatory elements near several WNT receptors (Frz1,5,7 & 8; Supplemental Figure 2 I).

Regarding WNT signaling, Faedo et al, Cerebral Cortex, 2007 provided evidence that loss of Nr2f1 increased the ventral spread of the bat-gal WNT reporter allele, whereas gain of Nr2f1 repressed Bat-gal activity.

-------------

Reviewer #3: Tseng et al present a nice body of work examining the role of COUP-TF1 in patterning the dorsal-ventral axis of the hippocampus. Their work adds to the literature by using both knock out and an enhanced expression of COUP-TF1, and they conclude that COUP-TF1 knock out leads to the expansion of the dorsal hippocampus (while reducing the ventral hippocampus) and that overexpression has the opposite effects (reduced dorsal hippocampus, expanded ventral hippocampus with ectopically expressed ventral hippocampal cells). I greatly appreciate their attention the different subregions of the hippocampus.

Because the curvature of the hippocampus can make it difficult to assess and align sections of a shortened or elongated hippocampus with that of wild-type mice, the analyses with Nissl-stained sections are harder to believe than the tissues that are stained with specific markers of cell type and location along the dorsal-ventral axis. In particular, the ventral morphology of the cKO is quite challenging to align with the other two genotypes and it is difficult to tell how much of that is due to cutting this more caudal tissue at a very different angle due to a shortened curvature. Perhaps a more appropriate prep would be to dissect the hippocampus and section the flattened hippocampus to compare dorsal, intermediate, and ventral across genotypes.

Similarly, scaling up the cKO and scaling down the cTG is also difficult to interpret because it isn't clear if the differences in sizes are evenly distributed, or how they relate to the control. Instead, it might be useful to do percent of the axis, and certainly the qRT-PCR analysis of the whole hippocampus is more compelling. 

Of note, the labeling of vCA1 in the cKO image of Figure 2A is questionable. The authors have identified a window of very thin pyramidal layer to measure and claim is thinned in the cKO mouse, but they have labeled the CA1 layer quite extensively into what is likely the CA3 layer, based on the widening of the cells. Clearer delineations of the subfields would give the reader more confidence in the assessment of the subregions by this group. 

I would like to note a few issues in the discussion: 

"In this study, we showed that even at an early postnatal stage (P7), when not all the subregion-specific genes are fully expressed, the Hp can still be divided into dorsal and ventral compartments using a handful of marker genes. This suggests that Hp compartmentalization takes place early in development, well before the Hp begins to show adult-like functions at three to four weeks after birth [39]."

While this is true, they do not note that both connectivity of the hippocampus with its main cortical input the entorhinal cortex, as well as dorsally or ventrally enriched gene markers at the dorsal/ventral extremes of the neonatal hippocampus are identifiable at birth (O'Reilly et al Brain Structure and Function 2015).

Additionally, they state: "Further compartmentalization and refinement of these subregions likely takes place during Hp maturation, as physical activities are known to significantly contribute to hippocampal development in humans [40, 41]."

Isn't "further compartmentalization and refinement" the definition of maturation? This sentence feels empty and vague and the references 40 and 41 are taxi driver studies; are these two studies really support for the role of "physical activities to significantly contribute to hippocampal development in humans?" Instead the authors could cite articles that indicate the sensory input is relevant for hippocampus maturation, such as studies of early eyelid opening leading to earlier maturation of the dentate gyrus (e.g. Dumas Dev Psychobiology 2004), or perhaps relate the timing of their studies to other changes in maturation across sensory systems, because sensory input is important for hippocampal-based learning an memory. 

Finally, they state that the hippocampus has been functionally described to have gradients along the axis but that "when analyzed at the molecular level, the hippocampus can be divided into distinct domains. Different sets of genes are enriched in specific subregions of the adult Hp, including the dHp, vHp, and an intermediate region between them [10, 38]." 

Citation 38 by Fanselow presented the case that the two domains may not be distinct, based on gradients of molecular labeling. If you look at the extremes, you can identify distinctions, but the boundaries are not necessarily clear cut. 

A few minor comments:

1) The use of dHp and vHp and then swapping back and forth between this nomenclature and the rostral-caudal nomenclature makes the manuscript harder to follow. 

2) in Figure 2, it is very hard to see the letters a1-a6 on the images.

---------------

Reviewer #4: 

In their manuscript entitled "Determining the relative sizes of hippocampal subregions along the dorsal-ventral axis by COUP-TFI expression gradient", Tseng and colleagues highlight the role of transcriptional regulator Couptf1 in the regionalization of the hippocampus and investigate how the expression gradient hippocampal progenitors (high-caudal to low-rostral) may contribute to distinguishing the fate of dorsal and ventral hippocampal regions. The authors used Couptf1 conditional knockout (cKO) and Coutf1 overexpression (cTG) models to study the effects of decreased or increased Couptf1 expression in the emergence of dorsal and ventral hippocampal fate and in the molecular regionalization of the hippocampus. 

Using anatomical analysis and expression of molecular markers they proposed that Couptf1 deletion results in the expansion of the dorsal hippocampus at the expense of the ventral region, and that increased Couptf1 expression results in ectopic generation of ventral hippocampus in the dorsal region. Moreover, they point to Wnt signaling as a mechanism regulated by Couptf1 to control dorsoventral regionalization in the hippocampus.

Overall, Tseng et al. highlight an important concept linking Couptf1 to the regionalization of the hippocampus. The involvement of Couptf1 in hippocampal regionalization has been proposed previously, and this study supports Couptf1's role in hippocampal morphogenesis. This study offers a new perspective on how Couptf1 may regulate hippocampal regionalization, contrasting the current view in the literature. However, the study presents important shortcomings in supporting their main claims. 

Main concerns:

1. One of the author's main claims is that Couptf1 regulates the dorso-ventral balance of CA1, as well as the position of CA2. The authors claim that the dorsal hippocampus (dHP) expands at the expense of the ventral hippocampus (vHP) in the absence of Couptf1. Previous publications (cited by the authors) using two different Couptf1 mutant models (including the Emx-Cre;Couptf1fl/fl system used in this study) have shown that Couptf1 is necessary for the specification and differentiation of the dorsal hippocampus, and its mutation mostly affects the differentiation of the dorsal CA1. This is in stark contrast with the author's main claims however the authors do not discuss this discrepancy.

The authors show that the HP is smaller in the Couptf1 cKO (as observed in previous studies), and that the thickness of the pyramidal layers in CA1 and CA3 changes. These morphological changes indicate the abnormal development of these fields. The study argues that the reduced thickness and increased cell density in the cKO CA1 and the increased thickness and decreased cell density in the cTG CA1 suggested that the Hp was "dorsalized" in the cKO and "ventralized" in cTG. These anatomical changes are unlikely to reliably distinguish regional differences given the dysmorphism of the HP in the cKO. Their claim that dHP expands at the expense of the vHP is mostly supported by the expanded expression of the Wsf1 marker in CA1 from rostral to caudal (Figure 2D). Of note, expression of Wsf1 in dHP CA1was shown in the study of Yang and colleagues (doi: 10.7554/eLife.86940) in the same Couptf1 cKO model (Emx1Cre; Couptf1 fl/fl), and appeared to be reduced rather than expanded. There is no discussion regarding the choice of the markers, and given the discrepancy with the literature, it diminished the strength of the data yielded by markers that may not be reproducible across studies.

The upregulation of other dHP markers by qPCR from whole HP in the cKO is not very informative, since it may also reflect an overall abnormal differentiation and misregulation of these genes across the HP rather than the specific expansion of the dHP. Showing upregulation and expanded expression of multiple dHP markers in the tissue via spatial transcriptomics or in situ hybridization will better support this point.

Given the discrepancies in Wsf1 expression with the literature, the claim regarding the "expansion of dHP" needs to be further supported by showing expression in situ of additional markers that distinguish dHP and vHP. Perhaps using the same markers as in the previous studies will help to clarify the starkly different results across studies, in addition to the use of new markers which will add to the rigor and strength of the data.

Similarly, the support for the claim that the vHP is ventrally shifted concomitant with the expansion of the dHP is unclear. Their claim that Dcn (vHP marker) expression is shifted ventrally is unclear based on the images in Figure 2D, and not supported by Dcn quantification in Figure 2F, which does not show statistically significant differences in Dcn between the cKO and the control. The authors showed reduced expression of other vHP makers by qPCR from the whole HP, which as explained above, only indicates an overall reduction in the expression, but is not useful to support the claim that the vHP has shifted ventrally, has become smaller, or any positional effect. 

Overall, the authors are pushing the conclusion that the balance between the regions specified as dHP and vHP is shifted in the Couptf1 cKO favoring the expansion of dHP. Though the premise is interesting, the data presented are currently insufficient to support that conclusion. Stronger data (showing the expansion of expression in situ of several makers that unequivocally distinguish dHP and vHP) is needed to claim that the territory demarcated as dHP has acquired the correct molecular identity of dHP and that the region is indeed expanded.

In addition to the examination of molecular markers distinguishing dHP and vHP, the authors mentioned the distinct connectivity of dHP and vHP with other structures such as the cortex, amygdala, hypothalamus, etc. The authors could use retrograde tracing from different structures to identify the dHP and vHP regions, and determine their identity based on connectivity as an additional method.

2.Location of CA2, and boundary between CA1, CA3 doesn't seem to change in Couptf1 cKO compared to control.

The authors show that the expression of Pcp4 (CA2 marker) is detected between CA1 and CA3, similarly in cKO and controls (Figure 3A), and state "In the cKO, a Pcp4+ domain was detected between CA1 and CA3, similar to the control". If the relative position of CA2 is similar in Couptf1 cKO and controls, how do the authors reconcile these data with their conclusion regarding the dorsalization of CA1 and the expansion of the dCA1? Shouldn't CA2 be affected/shifted too?

3. Couptf1 regulation of Wnt downstream genes. This is a potentially interesting mechanism revealed by the authors' ChIP-seq data. They suggest that Axin2 and Lef1 expression is regulated by Couptf1 in the neuroepithelium. In support of this claim, they show the expression of Axin2 and Lef1 by in situ hybridization. Images illustrating changes in Lef1 expression level between cKO, cTG, and controls are convincing, though it is unclear how the authors quantified protein expression level in Figure 4D, as claimed in the text. The images presented in Figure 4C illustrating Axin2 expression changes are less convincing and there is no quantification of Axin2 expression. Clarification on the quantification of expression, and perhaps additional data supporting these changes in expression would strengthen this interesting claim.

Minor comments:

1. Incomplete header: "COUP-TFI expression levels regulate the position of CA2 and the dorsal-ventral balance in CA3 and"

2. The statistical tests applied in each figure showing quantitative comparisons are not described in the figure legends.

3. The method lacks detailed description and rigor. There is no methodological description for "protein quantification" for Couptf1 and Lef1 described in the text corresponding to experiments presented in Figure 4D. What kind of protein quantification method was performed? 

The 'Stereological analysis, Image acquisition, and Quantification' methods section is poorly described. It seems from their description that the authors performed their volumetric analysis using planimetry, not stereology. If stereology was indeed used, they need to state the probe they used for area estimation and provide the error coefficient for their estimation. It is important to state at which magnification the images were acquired. This influences the accuracy of measurements and quantification.

---

## [Decision Letter · Decision Letter 2]

23 Jul 2025

*Dear Dr Chou,

Thank you for your patience while we considered your revised manuscript "Determining the relative sizes of subregions of the hippocampus proper along the dorsal-ventral axis by COUP-TFI expression gradient" for publication as a Short Report at PLOS Biology. This revised version of your manuscript has been evaluated by the PLOS Biology editors, the Academic Editor and several of the original reviewers.

Based on the reviews and our Academic Editor's assessment of your revision, we are likely to accept this manuscript for publication, provided you satisfactorily address the following data and other policy-related requests. Please also see the final comment from the Academic Editor, following the reviewer reports.

IMPORTANT: the following editorial points are listed point-by-point (**)

------------

**Title: 

-- We suggest the following alternative title: "A developmental gradient of COUP-TFI expression regulates the relative size of hippocampus dorsal and ventral subregions"

**Financial disclosure statement:

Please add links to the funding agencies in the Financial Disclosure statement in the manuscript details.

**Competing interests: 

Please modify your Competing Interests the following statement “YPH is a member of PLOS Biology’s Editorial Board. The other authors declare that no competing interests exist."

**Ethics: 

-- Please include the specific national or international regulations/guidelines to which your animal care and use protocol adhered. Please note that institutional or accreditation organization guidelines (such as AAALAC) do not meet this requirement.

**Data:

Please supply the numerical values either in the a supplementary file or as a permanent DOI’d deposition for the following figures:

1EFGH

2BCFGH

3B

S1B

S2DE

S3ABCC'

S6C

-- Please cite the location of the data clearly in all relevant main and supplementary Figure legends, e.g. “The data underlying this Figure can be found in S1 Data” or “The data underlying this Figure can be found in https://doi.org/10.5281/zenodo.XXXXX”

-- Please ensure that you are using best practice for statistical reporting and data presentation. These are our guidelines https://journals.plos.org/plosbiology/s/best-practices-in-research-reporting#loc-statistical-reporting and a useful resource on data presentation https://journals.plos.org/plosbiology/article?id=10.1371/journal.pbio.1002128

-- If you are reporting experiments where n ≤ 5, please plot each individual data point.

-- Supplementary files (e.g., excel). Please ensure that all data files are uploaded as 'Supporting Information' and are invariably referred to (in the manuscript, figure legends, and the Description field when uploading your files) using the following format verbatim: S1 Data, S2 Data, etc. Multiple panels of a single or even several figures can be included as multiple sheets in one excel file that is saved using exactly the following convention: S1_Data.xlsx (using an underscore).

--Please ensure that your Data Statement in the submission system accurately describes where your data can be found and is in final format, as it will be published as written there.

**Code availability:

**Model system:

--Please note that per journal policy, the model system/species studied should be clearly stated in the abstract of your manuscript. 

------------

We expect to receive your revised manuscript within two weeks. 

*Published Peer Review History*

*Press*

Sincerely,

Taylor

Taylor Hart, PhD, 

Associate Editor

thart@plos.org

PLOS Biology

Reviewer remarks:

Reviewer #1: I appreciate the time and care the authors have spent in addressing the comments. 

I am happy to endorse the manuscript for publication

Reviewer #3: All of my concerns were well-addressed.

Reviewer #4: The authors have satisfactory improved the manuscript. All points has been addressed either with addition of new data, new analysis, or editorial changes.

The resulting manuscript is stronger, suitable for publication in PLOS Biology, and will contribute to understand the role of Couptf1 in the regionalization of the hippocampus.

Comment from the Academic Editor:

Tseng et al. present a comprehensive study investigating the role of COUP-TF1 in regulating the regionalization of the mouse hippocampus. Using both loss- and gain-of-function COUP-TF1 mutants, combined with detailed anatomical and molecular analyses, they provide new insights into how COUP-TF1 influences hippocampal patterning. This work offers a novel perspective that broadens our understanding of hippocampal regionalization.

---

## [Editor Report · Decision Letter 3]

5 Aug 2025

Dear Dr Chou,

Thank you for completing our editorial requests, including providing source data for your figures. However, we noticed some remaining issues related to the source data, and request that you address them through another Minor Revision.

The data listed for Figure 1F appear to be inconsistent with what's shown in the relevant figure, as it appears that the columns are mismatched. In addition, we noticed that the same excel sheet contains some additional numbers on the right side of the page with uncertain meaning. In your next revision, you should address these issues, and also ensure that all of the provided data is correct and matches the figures. Please also add source data for Figure 1I, which we missed in our prior request.

We expect to receive your revised manuscript within two weeks. 

*Published Peer Review History*

*Press*

Sincerely,

Taylor

Taylor Hart, PhD, 

Associate Editor

thart@plos.org

PLOS Biology

---

## [Editor Report · Decision Letter 4]

8 Aug 2025

Dear Dr Chou,

Thank you for the submission of your revised Short Report "A developmental gradient of COUP-TFI expression regulates the relative size of hippocampus dorsal and ventral subregions" for publication in PLOS Biology. On behalf of my colleagues and the Academic Editor, Maria Galazo, I am pleased to say that we can in principle accept your manuscript for publication, provided you address any remaining formatting and reporting issues. These will be detailed in an email you should receive within 2-3 business days from our colleagues in the journal operations team; no action is required from you until then. Please note that we will not be able to formally accept your manuscript and schedule it for publication until you have completed any requested changes.

PRESS

Sincerely, 

Taylor Hart, PhD, 

Associate Editor

PLOS Biology

thart@plos.org